# The Effects of the Addition of Ti₃SiC₂ on the Microstructure and Properties of Laser Cladding Composite Coatings

**Qin Shi** [1,2], **Hejun Zhu** [1] **and Changsheng Li** [1,2,*]

[1] School of Mechanical Engineering, Zhenjiang Vocational Technical College, Shaoxing 312099, China; ljftxzq@163.com (Q.S.); hehe666777@163.com (H.Z.)

[2] School of Mechanical Engineering, Jiangsu University, 301, Xuefu Road, Zhenjiang 212013, China

[*] Correspondence: lichangsheng@ujs.edu.cn

**Abstract:** This study explored the effects of Ti₃SiC₂ on the microstructure and properties of laser cladding coatings using X-ray diffractometer, scanning electron microscope, electrochemical workstation, and UMT-2 wear tester analyses. It was found that with the addition of Ti₃SiC₂, the reinforcing phases in the composite coating were TiC, Ti(B,C)₂, honeycomb-like (Cr, Fe)₂₃C₆, and a novel composite ceramic with an "eyeball" structure, which had an inside core of Al₂O₃ and TiC outer surrounding structure. The microhardness, wear, and corrosion resistance of the composite coating were about 1.35, 2, and 4.3 times those of the original coating, respectively. The main wear mechanisms of the original coating were severe fatigue spalling and microcutting, while the main mechanisms of the composite coating were slight microcutting and the formation of the transferred film.

**Keywords:** Ti₃SiC₂; composite ceramics; wear and corrosion resistance

## 1. Introduction

Owing to its appropriate strength, tolerance of design stress, and low cost, 16Mn steel has been widely applied in industry, such as in transportation equipment, mining machines, and chemicals. In most cases, 16Mn steel is subjected to repeated impacts and corrosion wear in working conditions. Hence, in order to resist severe wear and increase its service life, its weaknesses, including its low hardness and poor resistance to wear, need to be improved. As the friction and wear only occur on the surfaces of components, surface strengthening for 16Mn steels is a more cost-effective approach. Conventional surface treatment techniques, such as thermal spraying [1], cold spraying [2], physical vapor deposition [3], and chemical vapor deposition [4], have been reported to be feasible methods of improving the surface properties of the steel substrate by depositing high-performance coatings on its surfaces. Additionally, these coatings were also shown to have high hardness and excellent corrosion and wear resistance, but they are susceptible to being detached from the substrate in aggressive wear working conditions because of the mechanical bonding between them and the substrate. Laser cladding (LC) employs a laser as the heat source to fabricate hard protective coatings on steel substrates [5–7]. Additionally, compared with the above surface strengthening techniques, laser cladding confers the unique advantages of a dense and refined microstructure, little stress deformation, a small heat effect zone, and firm bonding. As such, it is regarded as one of the most optimal surface techniques [8–10].

In the field of laser cladding, a number of advances for coatings with various additives have been reported. Some ceramics, such as B₄C, ZrB₂, and tungsten carbide have been directly added to cladding material systems, which were then heated by a laser beam and dissolved into a melt pool, forming diverse new hard phases in composite coatings in situ [11–13]. Cr, Co, Ta, and some

special metal elements could also be added into the alloy powder, which could effectively improve the microstructure of the alloy coating, decrease the crack susceptibility, and increase the resistance to corrosion and wear [14–17]. In addition to the above ceramics and metals, some researchers have also found that rare earth oxides ($Y_2O_3$, $CeO_2$, $La_2O_3$, etc.) could refine the microstructure of laser cladding coatings and increase their microhardness, corrosion, and wear resistance [18–20]. $Ti_3SiC_2$, one of the typical ternary layered compounds, which has the combined advantages of both metals and ceramics, has attracted much attention [21]. Additionally, $Ti_3SiC_2$ was also reported to have good lubricating properties [22,23]. Considering the abovementioned advantages, $Ti_3SiC_2$ could become a promising reinforcement for laser cladding coatings. However, little research on $Ti_3SiC_2$ reinforced laser cladding coatings has been published to date.

In the present article, an investigation of the effects of $Ti_3SiC_2$ on the microstructure and properties of laser cladding coatings was carried out in detail. This was expected that the design of better wear-resistant coatings would be exclusively used in industry.

## 2. Experimental Procedures

### 2.1. Preparation of Coatings

In this study, 16Mn steel measuring 100 mm × 40 mm × 8 mm was adopted for the steel substrate. The uniform mixture of Fe60 alloy and $Ti_3SiC_2$ particles was obtained using a planetary ball mill, and the content values of $Ti_3SiC_2$ particles were 0 and 10 wt.%, respectively. Table 1 lists the chemical compositions of the steel substrate and Fe60 alloy powders. Figure 1 shows the morphologies of Fe60 alloy and $Ti_3SiC_2$ powders. The average size of Fe60 alloy powders was about 100 μm, while $Ti_3SiC_2$ particles were flaky, with an average size of about 20 μm. Before laser cladding, the substrate surface was ground with SiC sandpaper. After mixing the coating materials with the water glass solution ($Na_2O \cdot nSiO_2/H_2O$ = 1:3, vol.%), a powder bed slurry was formed on the surface of the substrate with a thickness of about 1.0 mm, which was dried at a temperature of 333 K for 5 h.

**Table 1.** The chemical compositions of the 16Mn substrate and Fe60 alloy powder (in wt.%), measured by energy dispersive spectrometer (EDS).

| Material | Fe | Mn | Ni | Cr | C | Si | Cu | P | S | B |
|---|---|---|---|---|---|---|---|---|---|---|
| 16Mn | Bal. | 1.2~1.6 | ≤0.3 | ≤0.3 | 0.1~0.2 | 0.2~0.6 | ≤0.25 | ≤0.03 | ≤0.03 | – |
| Fe60 | Bal. | – | 0.1~1 | 13~17 | 0.5~1.0 | 0.3~1.0 | – | – | – | 0.2~1.5 |

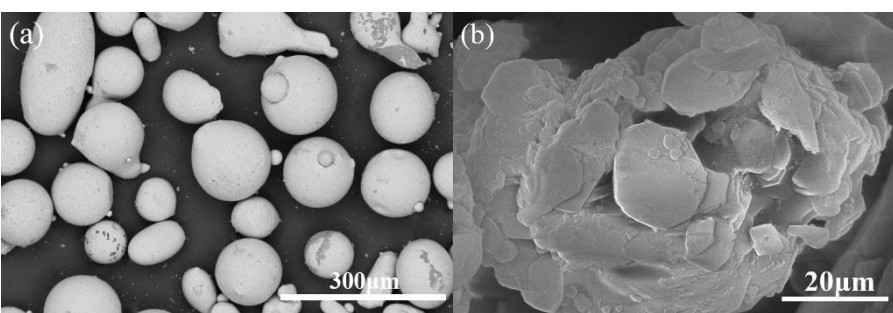

**Figure 1.** The morphologies of experimental powders: (**a**) Fe60 alloy powder; (**b**) $Ti_3SiC_2$ particles.

Next, a Laserline laser cladding system (LDF4000-100, Redmond, WA, USA), a high power continuous-wave semiconductor laser with a wavelength of 980 nm, was employed for cladding experiments, whereby the laser beam was adjusted into a rectangle laser spot measuring 12 mm × 3 mm by a 100 mm optical lens. Table 2 lists the details of the suitable laser processing parameters. During the experiment, argon was passed into the molten pool continuously, providing a protective atmosphere

to protect the melt alloy from being oxidized. The macro morphologies of as-prepared coatings are shown in Figure 2.

**Table 2.** The suitable process parameters and chemical compositions of preplaced powders.

| Samples | Composition (wt.%) | Power (wk) | Scanning Speed (mm/s) |
|---|---|---|---|
| Coating 1 | Fe60 | 4.0 | 6 |
| Coating 2 | Fe60/10Ti$_3$SiC$_2$ | | |

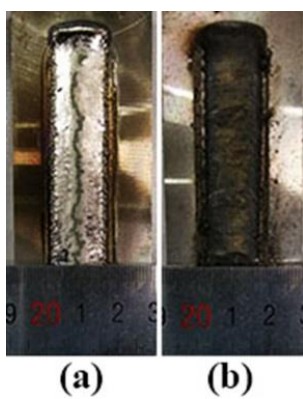

**Figure 2.** Surface morphologies of laser clad coatings: (**a**) Fe60; (**b**) Fe60/10 wt.%Ti$_3$SiC$_2$.

*2.2. Microstructure Characterization and Property Measurements*

The cross-sections of cladded coatings were abraded, polished, and finally chemically etched with a mixed solution of alcohol-HNO$_3$ (5 vol.%) for around 0.5 min. The phase constituent and microstructure were identified with the assistance of a X-ray diffractometer (XRD-6000, Shanghai Rente Testing Instrument Co., Ltd., Shanghai, China), scanning electron microscope (Sigma, ZEISS Merlin Compact, Oberkochen, Germany), and energy dispersive spectrometer (EDS, ZEISS, Oberkochen, Germany).

The microhardness distribution in the cross-section of the composite coatings was measured via a Vickers hardness tester (THV-5MD) (Lab Testing Technology (Shanghai) Co., Ltd., Shanghai, China) with a 0.3 Kg test load and 10 s holding time. Corrosion experiments were performed in two ways: potentiodynamic polarization and electrochemical impedance spectroscopy (EIS). The measurements were made using an Autolab system (Model Aut84886) (Shanghai Chenhua Instrument Co., Ltd., Shanghai, China), which connected to a three-electrode electrochemical cell. The polarization curves and electrochemical impedance spectroscopy of the composite coatings were measured in 3.5 wt.% NaCl solution at room temperature.

The dry wear tests were done using a "ball-on-disk" UMT-2 wear tester (CETR, Bruker, Billerica, MA, USA). An Al$_2$O$_3$ ceramic ball with a diameter of 10 mm and a hardness greater than HRC95 was used as the counterpart. Composite coatings were used as specimens. The test load was 30N, the linear velocity was 10 mm/s, and the test time was 60 min. The wear mass loss after the sliding tests was measured at an interval of 15 min using a precision analytical balance with a minimum scale value of 0.1 mg.

## 3. Results and Discussion

*3.1. Macroscopic Appearance and Phase Composition*

Figure 2 shows macro-photographs of the two composite coatings. Coating 1 had a smooth and uniform surface appearance. In contrast, the surface of coating 2 was relatively coarse and covered with some spherical particles. This was because Ti$_3$SiC$_2$, with a density of approximately 4.53 g/cm$^3$, had a lower density than Fe60 alloy (approximately 7.8 g/cm$^3$), meaning partial Ti$_3$SiC$_2$ particles could

rapidly rise and splash from the molten pool in the cladding process, adhering to the coating surface to form spherical particles. Similarly, this phenomenon also took place in the reports by Liu et al. and Lu et al. [24]. Additionally, the surface color of coating 2 was far darker than that of coating 1, which was related to certain chemical reactions induced by laser heating [25].

Figure 3 shows the cross-sections of two composite coatings. As presented in Figure 3a,b the thickness of coating 2 was far smaller than that of coating 1. This is because the addition of $Ti_3SiC_2$ led to powders splashing on the laser cladding, which decreased the dilution. It was also observed that both clad coatings were quite dense, and there were several holes and cracks visible in the junctures between clad coatings and the substrate. Moreover, the melted alloy atoms at the interfaces between the clad coatings and the substrate appeared to show mutual diffusion, as clearly shown in Figure 4. This above evidence suggests that two clad coatings adhered well to the substrate [26]. In addition, the acicular martensites were obviously produced near the clad coating-substrate interfaces, as indicated in Figure 3c,d, which was attributed to the rapid heating and cooling rate [27,28].

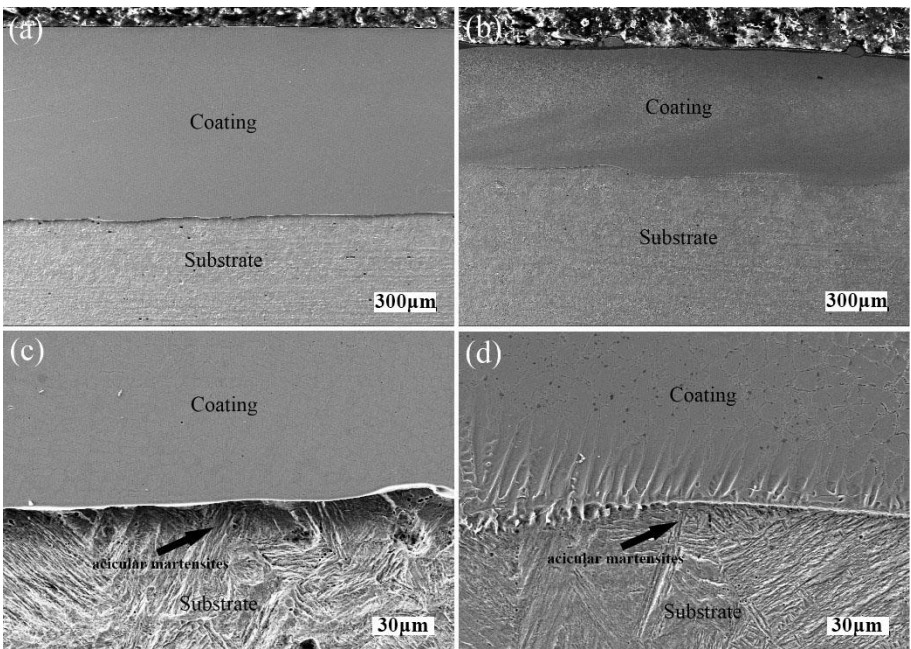

**Figure 3.** Macrograph of transverse cross-sections of Coating 1 and Coating 2: (**a**) overview of coating 1; (**b**) overview of coating 2; (**c**) bonding area of coating 1; (**d**) bonding area of Coating 2.

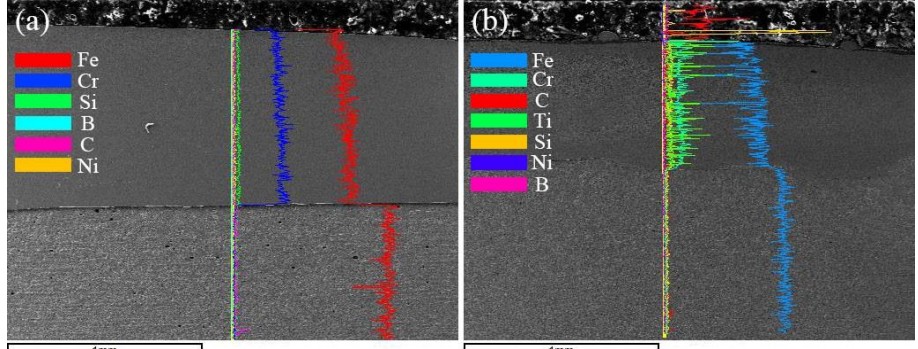

**Figure 4.** Linear elemental distributions of (**a**) Coating 1 and (**b**) Coating 2.

To examine the phase constitution of coatings, their XRD spectra are presented in Figure 5a. After analyzing the diffraction peaks, $\alpha$-Fe, $\gamma$-(Fe, Ni), (Cr, Fe)$_{23}$C$_6$, CrB, and Fe$_3$C were the main phases of the two composite coatings. Partial Cr atoms in Cr$_{23}$C$_6$ were replaced by Fe atoms to form

(Cr, Fe)$_{23}$C$_6$, which was supported by the fact that the atomic radius of Fe was close to that of Cr and their positions in the periodic table were very close [29]. Compared with coating 1, coating 2 also detected some additional diffraction peaks, which belonged to TiC, TiB$_2$, and Al$_2$O$_3$. This was because Ti$_3$SiC$_2$ particles dissolved in the molten pool to decompose Ti, Si, C, and Al atoms. In this molten pool, Ti, C, Al, O, and B combined with each other to generate TiC, TiB$_2$, and Al$_2$O$_3$. In addition, the diffraction peaks of Ti$_3$SiC$_2$ were also detected in coating 2, which demonstrated that a small amount of Ti$_3$SiC$_2$ was not dissolved in the molten pool and was reserved in coating 2 after cladding. Because of its hexagonal layered structure, residual Ti$_3$SiC$_2$ would contribute to the friction-reducing property of the alloy coating during the sliding process. Interestingly, the diffraction peaks of Ti-Si compounds were not present in the diffraction patterns. From a thermodynamic point of view, due to the high affinity between C and Ti, TiC tended to form before Ti-Si compounds [11]. TiC formation consumed a lot of Ti atoms, leading to Ti depletion in the alloy system, meaning Ti-Si compounds were not synthesized [30]. Additionally, the formation of (Cr, Fe)$_{23}$C$_6$ directly caused the loss of Cr atoms, which increased the tendency of Fe to form cementite (Fe$_3$C). It is obvious that diffraction peaks for γ-(Fe, Ni) showed a lower relative intensity in coating 2 than in coating 1, indicating that the quantity of γ-(Fe, Ni) was lower in coating 2 than in coating 1. Some previous research studies [31,32] reported that the martensite start temperature (Ms point) decreased with the increment of the content of C elements in austenite during the laser cladding process. Original Ti$_3$SiC$_2$ absorbed a lot of heat but the irradiation of the laser beam was dissolved into the molten pool, which released free Ti atoms. Cr and Ti atoms form Cr$_{23}$C$_6$ and TiC with element C, thereby reducing C atoms dissolved in austenite, in turn reducing the chemical stability of γ-(Fe, Ni) in coating 2.

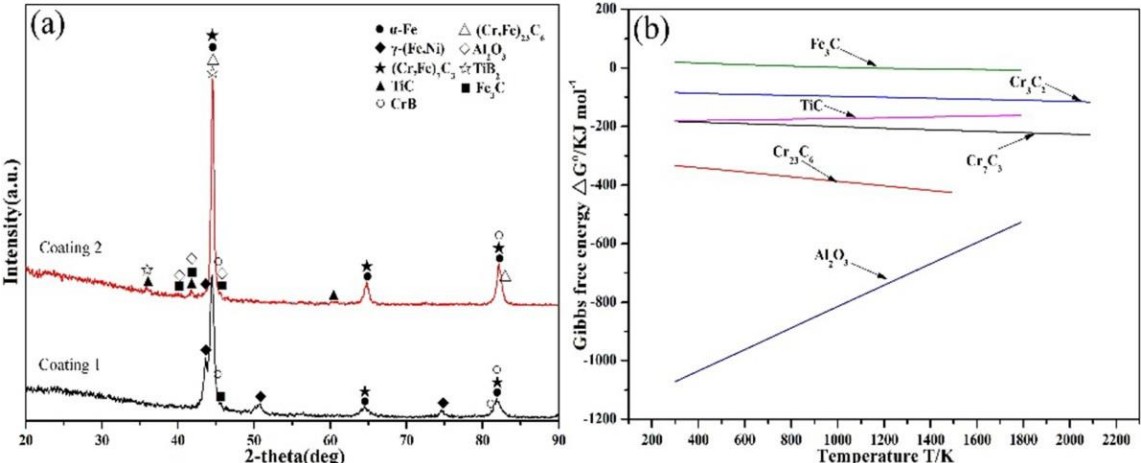

**Figure 5.** X-ray diffraction (XRD) patterns of the composite coatings (**a**) and the Ellingham diagram of Gibbs free energy via the reaction temperature (**b**).

The Gibbs free energy values (ΔG$^0$) of hard phases were calculated, as shown in Figure 5b. Except Fe$_3$C, Gibbs free energy values (ΔG$^0$) of other phases were negative, as the reaction temperature ranged from 298 to 2000 K. Hence, Cr$_{23}$C$_6$, TiC, and Al$_2$O$_3$ could form spontaneously.

### 3.2. Microstructure Analysis

The microstructure of coating 1 is shown in Figure 6. Two distinct feature phases were clearly observed, which were cellular dendrites and eutectics, respectively. EDS test results (see Table 3) indicated that the content of Fe was overwhelmingly high in cellular dendrites, in which relatively low contents of Si and Cr are dissolved. However, eutectics consisted of relatively high concentrations of Cr. XRD analysis and EDS elemental analysis were combined to draw the conclusion that cellular dendrites and eutectics were supersaturated solid solutions of α-Fe, with small amounts of Si and Cr elements and γ-(Fe, Ni)/(Cr, Fe)$_{23}$C$_6$/CrB, respectively.

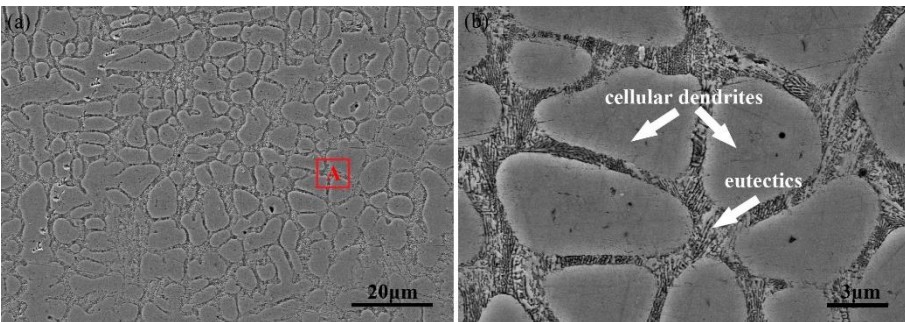

**Figure 6.** Scanning electron micrograph (SEM) image of Coating 1: (**a**) typical structure of the upper region and (**b**) the magnified morphology of the red rectangle A.

**Table 3.** Chemical composition of different phases examined by energy dispersive spectrometer (EDS) (wt.%).

| Phase | Fe | Ni | Cr | C | Si |
|---|---|---|---|---|---|
| Cellular dendrites | 52.47 | 1.12 | 12.88 | 31.64 | 1.89 |
| Eutectics | 52.30 | 1.24 | 24.06 | 21.39 | 1.0 |

Figure 7 shows the microstructural characteristics of the bottom, intermediate, and upper areas of coating 2. It is worth noting that the intermediate area showed a big difference in microstructure compared with the bottom area, as did the upper area. Here, $\alpha$-Fe and $\gamma$-(Fe, Ni)/(Cr, Fe)$_{23}$C$_6$/CrB eutectics were found in the bottom region. Eyeball-like phases (see Figure 8a) also accumulated around the grain boundaries. Generally, in situ formed particles would distribute at defect sites to decrease interfacial energy around grain boundaries, which was also mentioned in the study by Lei et al. [33]. EDS linear scanning was conducted to reveal the chemical composition of eyeball-like phases, as shown in Figure 9. As can be seen, Ti and C elements were mainly concentrated in the outer surrounding area, while Al and O elements were only prevalent in the core. Analysis with reference to XRD results indicated that the outer surrounding phase and the core were determined as TiC and Al$_2$O$_3$, respectively. Element valence analysis of the composite coating was done by X-ray photoelectron spectroscopy (XPS), as shown in Figure 10. The Ti 2p spectrum contained a wide peak, which had a binding energy of about 454 eV, which is characteristic of Ti-C bonds in TiC. The C 1s spectrum also contained a peak at about 284.5 eV, which might be associated with TiC. The O 2p spectrum contained a wide peak that had a binding energy of about 531 eV, which is characteristic of the Al–O bonds in Al$_2$O$_3$. The Al 2*p* spectrum also contained a peak at about 74.2 eV, which might be associated with Al$_2$O$_3$. Thus, Al$_2$O$_3$ and TiC were detected in XPS spectra. Al$_2$O$_3$ had far lower Gibbs free energy ($\Delta G^0$) than TiC, as shown in Figure 5b. Thus, Al$_2$O$_3$ very likely precipitated prior to TiC from the melt pool. As soon as Al$_2$O$_3$ particles formed, TiC depended on Al$_2$O$_3$ to grow. Figure 11 shows a schematic diagram of nucleation and growth behaviors of in situ synthesized Al$_2$O$_3$/TiC composite ceramics.

The elemental distribution of irregular cubic blocks was conducted, as shown in Figure 12. Clearly, it can be seen that cubic blocks were rich in Ti and C elements (not including Cr, Fe, Ni, B and Si elements), thus irregular cubic blocks were confirmed as TiC carbides. Because TiC had a high melting point (about 3140 °C) and negative free energy ($\Delta G^0$) (see Figure 6b), TiC carbides could primarily precipitate from the liquid via the nucleation growth mechanism during the rapid cooling process. In addition, honeycomb-like phases were distributed along grain boundaries of $\alpha$-Fe cellular dendrites, similarly to a network, and hexagonal phases were mainly distributed in $\alpha$-Fe cellular dendrites, as shown in Figure 8c. As shown in Figure 13, honeycomb-like phases and hexagonal phases were rich in Cr, Fe, and C elements; and Ti, B, and C elements, respectively. Combined with the XRD results, honeycomb-like phases and hexagonal phases were identified as (Cr, Fe)$_{23}$C$_6$ and TiB$_2$, respectively. In the present XRD analysis, as shown in Table 4 the 2θ of (101) peak of the TiB$_2$ structure was about

44.52° which was slightly higher than the standard peak of $TiB_2$ (2θ = 44.437°, JCPDS file 35-0741). Therefore, there is no doubt that some C elements dissolved into $TiB_2$ to form $Ti(B,C)_2$, which is a solid solution formed between TiC and $TiB_2$. Additionally, $Fe_3C$ was also detected in the composite coating, however its presence was not easily observed with SEM images.

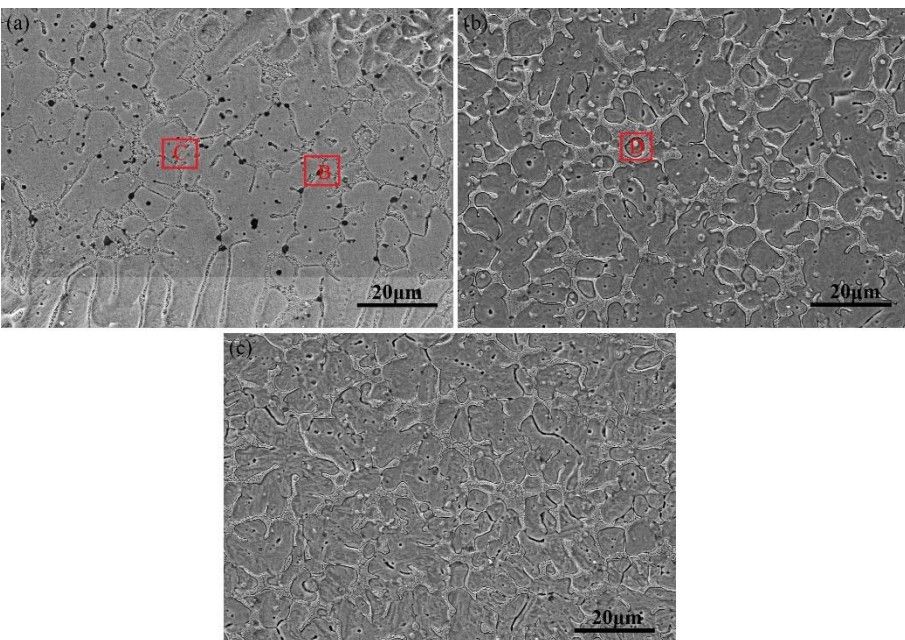

**Figure 7.** Scanning electron micrograph (SEM) image of coating 2: (**a**) bottom region; (**b**) intermediate region; (**c**) upper region.

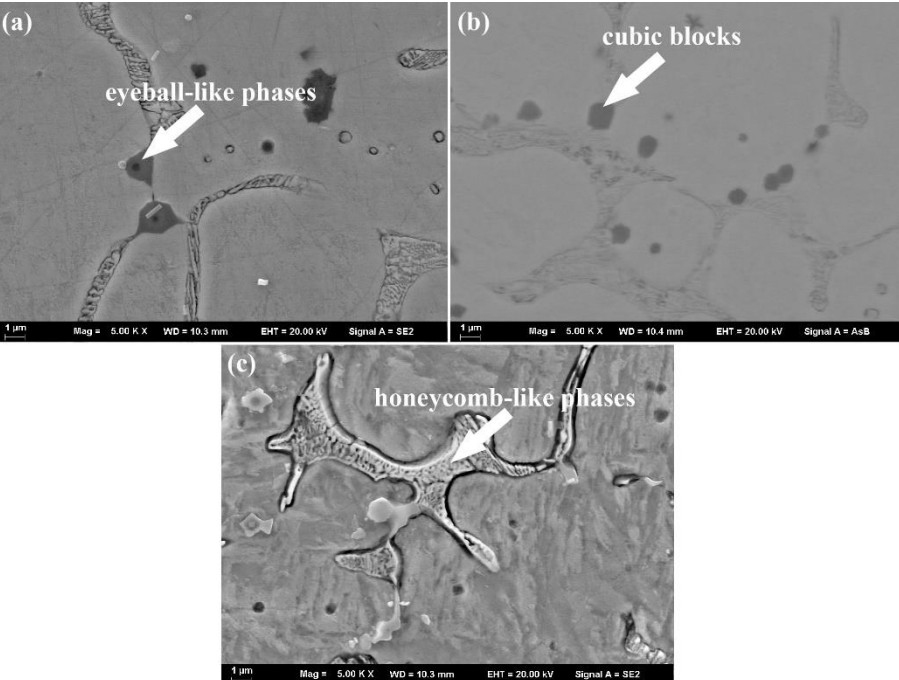

**Figure 8.** Enlarged views of red rectangles: (**a**) area B in Figure 7a; (**b**) area C in Figure 7a; (**c**) area D in Figure 7b.

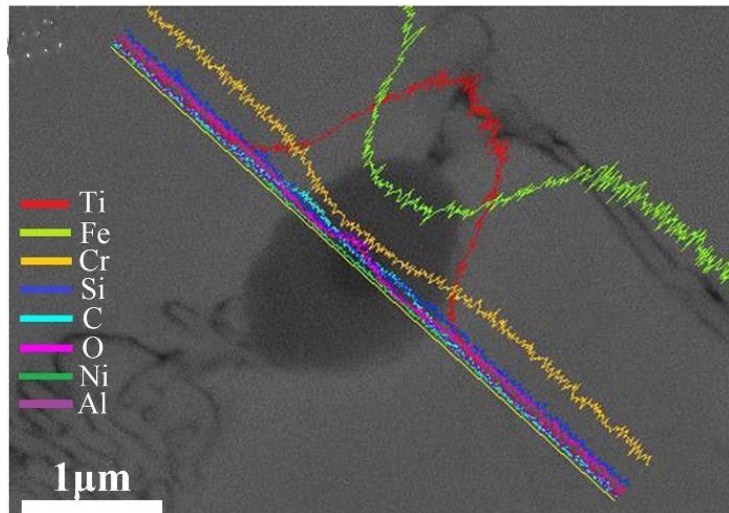

**Figure 9.** Energy dispersive spectroscopy (EDS) linear elemental distributions of eyeball-like phases in Figure 8a.

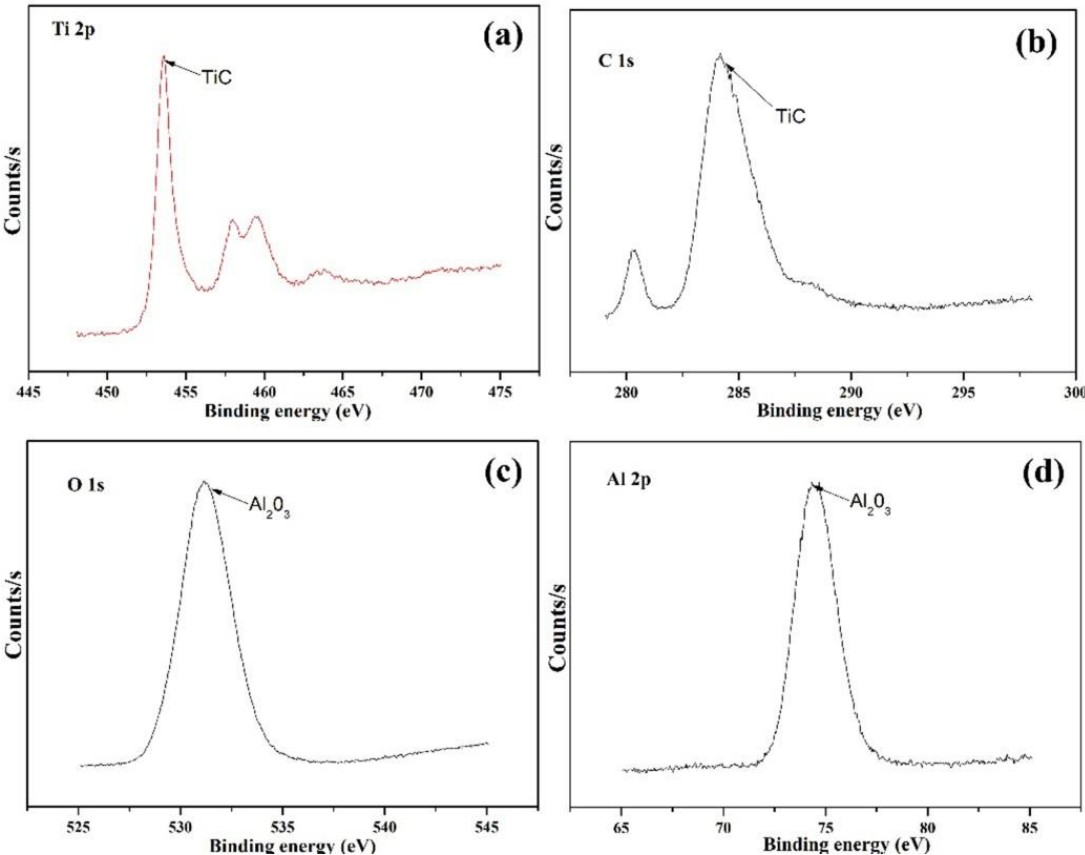

**Figure 10.** X-ray photoelectron spectroscopy (XPS) element valence diagrams of the composite coating with Ti$_3$SiC$_2$: (**a**) Ti 2*p*; (**b**) C 1*s*; (**c**) O 1*s*; (**d**) Al 2*p*.

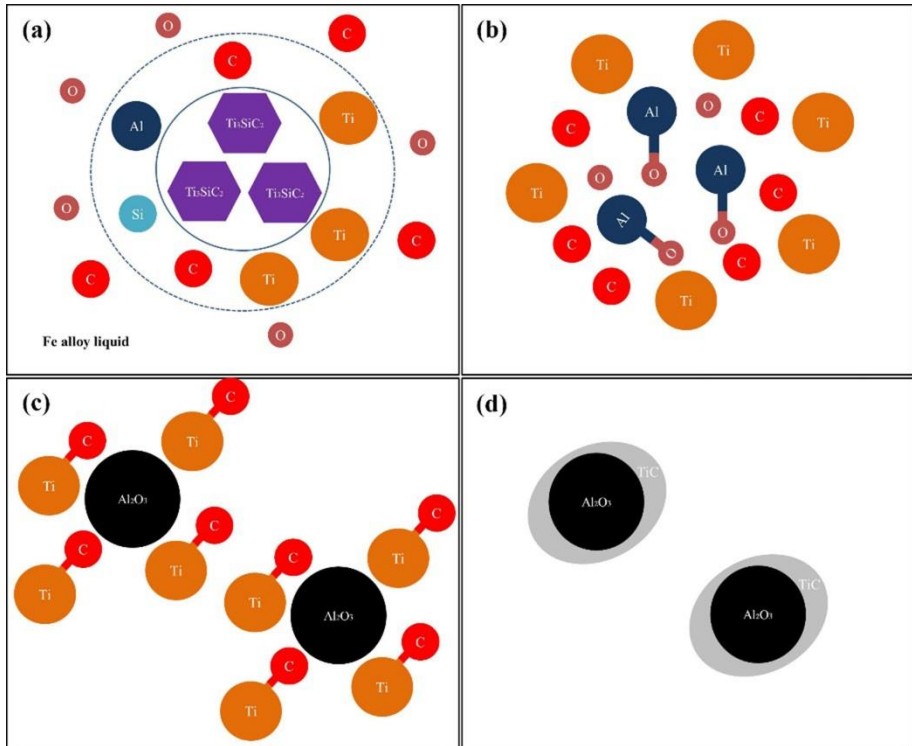

**Figure 11.** Schematic diagrams of nucleation and growth behaviors of Al$_2$O$_3$/TiC composite ceramics:
(**a**) dissolution of Ti$_3$SiC$_2$; (**b**) the bonding of Al-O; (**c**) the formation of Al$_2$O$_3$; (**d**) the formation of
Al$_2$O$_3$/TiC composite ceramics

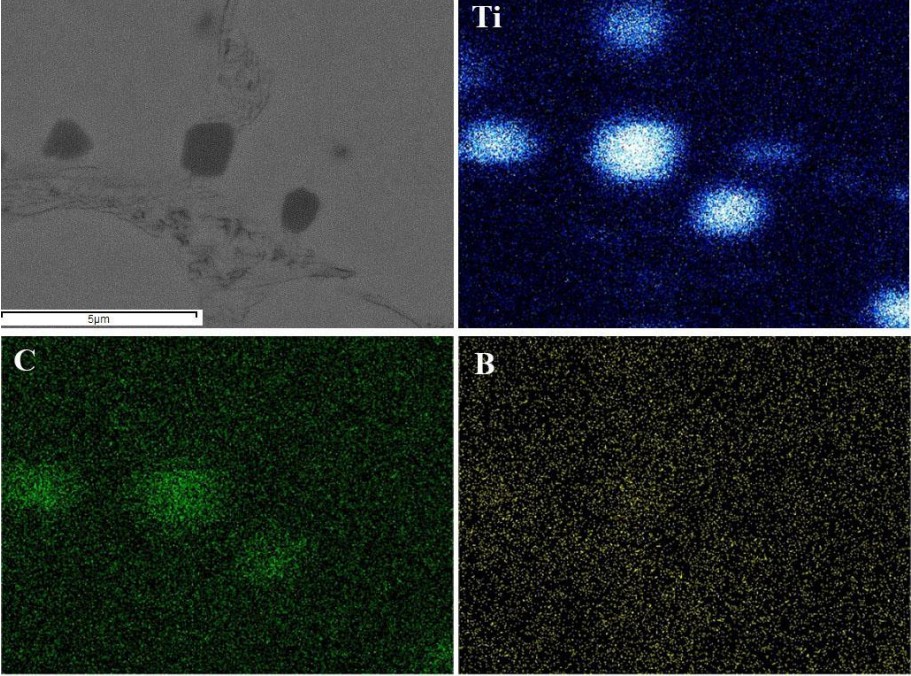

**Figure 12.** Elemental distributions of irregular cubic blocks in Figure 8b.

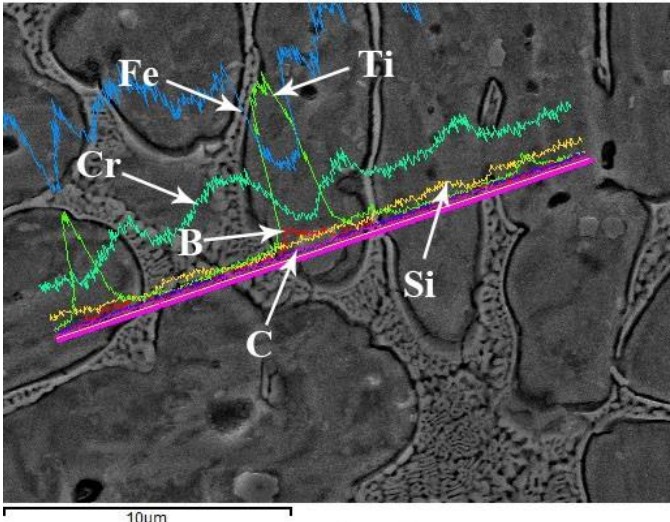

**Figure 13.** The elemental distributions of area H in Figure 10c.

**Table 4.** FWHM values of the three primary peaks in coating 1 and coating 2 ($2\theta/°$).

| Samples | Peak I | Peak II | Peak III |
|---------|--------|---------|----------|
| Coating 1 | 0.614 | F1.016 | 0.819 |
| Coating 2 | 0.558 | 0.614 | 0.636 |

### 3.3. Microhardness

Figure 14 shows the microhardness distribution as a function of distance from the surface of the clad coatings to the substrate. The average microhardness of coating 1 was about 554.8 $HV_{0.3}$. The microhardness distribution of coating 1 was uniform, which was mainly due to its uniform microstructure. The average microhardness of coating 2 was about 751.6 $HV_{0.3}$, which was about 1.5 times that of coating 1. The average microhardness of coating 2 was about 1.35 times that of coating 1, which was caused by the in situ formation of reinforcements, such as TiC, $Ti(B,C)_2$, $Fe_3C$, $(Cr, Fe)_{23}C_6$, and $TiC/Al_2O_3$ composite ceramics.

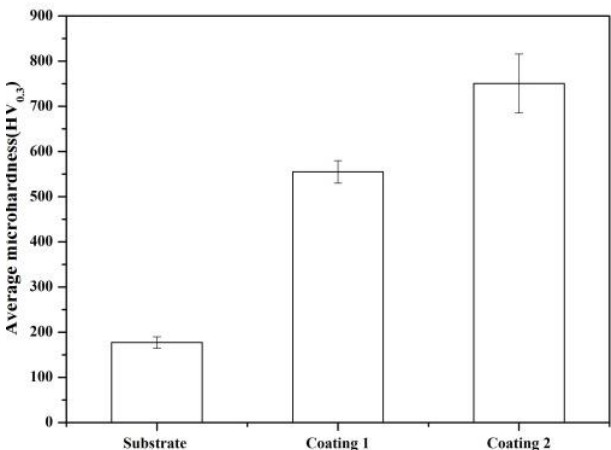

**Figure 14.** Microhardness profiles of transverse cross-sections of the composite coatings.

### 3.4. Electrochemical Properties

Figure 15a shows the potentiodynamic polarization curves of cladding layers in 3.5 wt.% NaCl solution. The passive phenomenon for coating 1 was not obvious, while an obvious passive phenomenon existed for coating 2. Table 5 lists parameters of the potentiodynamic polarization using traditional Tafel

methods. The corrosion current density ($I_{corr}$) of coating 1 (24.26 nA/cm$^2$) was an order of magnitude higher than that of coating 2 (5.631 nA/cm$^2$). The corrosion potential of coating 2 as compared with coating 1 shifted 0.257 V to positive, while the polarization resistance ($R_p$) of coating 2 (8923.2 Ω) was about 8 times higher than that of coating 1 (1079.6 Ω). Qualitative analysis showed that the corrosion potential ($E_{corr}$) shifted to a more positive state, the polarization resistance ($R_p$) increased, and the corrosion current density ($I_{corr}$) decreased, so the corrosion resistance of the cladding layers increased. However, quantitative analysis was also performed, which showed that the corrosion rate of coating 1 was 0.257 mm/a, which was about 6 times higher than that of coating 2 (0.0602 mm/a). According to some studies [31,32], the diameter of the capacitive impedance loops also decides the corrosion resistance of the cladding coatings. The larger diameter of the capacitive impedance loop is responsible for the better corrosion resistance. The EIS Nyquist curves of the composite coatings in 3.5 wt.% NaCl solution were measured, as shown in Figure 15b. It was observed that the diameter of the capacitive impedance loops of coating 2 was larger than that of coating 1. In addition, the ability to inhibit the penetration of electrolytes in the composite coatings was related to the middle frequency loop. The relationship between the middle frequency loop and the corrosion resistance had a positive correlation; that is, the larger the middle frequency loop, the better the corrosion resistance. Since the middle frequency loop of coating 2 was obviously larger than that of coating 1 in Bode curves, as shown in Figure 15c, the corrosion resistance of coating 2 was better than that of coating 1. This was also consistent with the study by Dai et al. [34]. Moreover, the $|Z|_{0.01}$ Hz values increased (Figure 15d) with the addition of Ti$_3$SiC$_2$, showing that the composite coating could serve as a more effective barrier layer against corrosion.

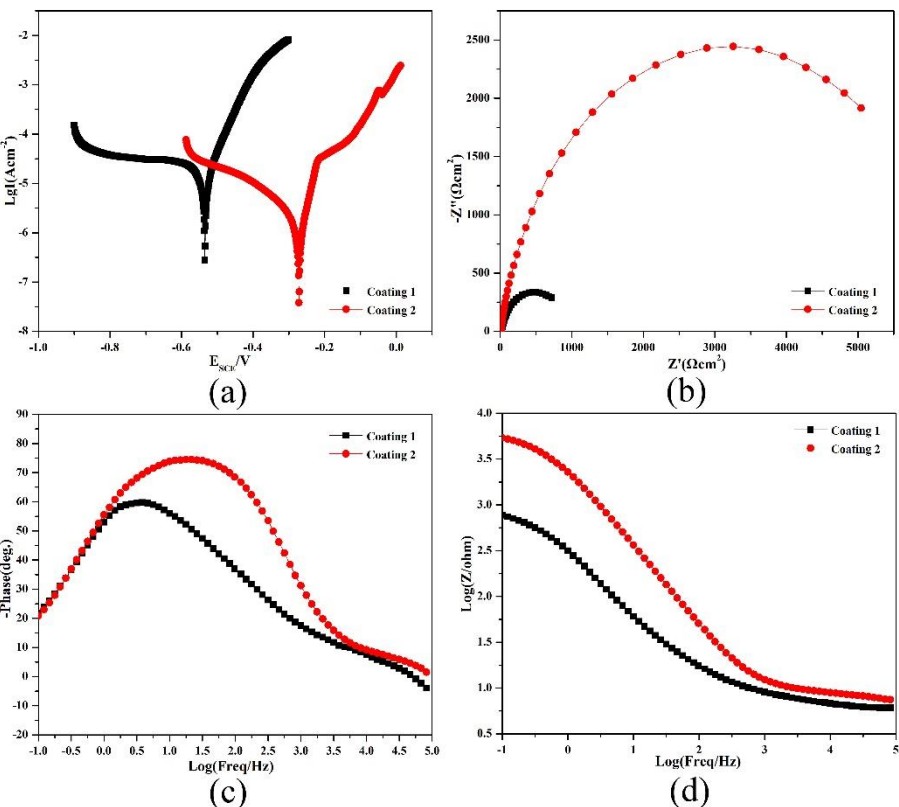

**Figure 15.** The curves of electrochemical tests: (**a**) anodic polarization curves; (**b**) Nyquist plots; (**c**,**d**) Bode plots.

**Table 5.** Fitting results of polarization curves.

| Samples | $I_{corr}$ (nA/cm$^2$) | $R_p$ ($\Omega$) | $E_{corr}$ (V) | Corrosion Rate (mm/a) |
|---------|------------------------|------------------|----------------|-----------------------|
| Coating 1 | 24.26 | 1079.6 | −0.461 | 0.257 |
| Coating 2 | 5.631 | 8923.2 | −0.311 | 0.0602 |

The EIS results further confirmed that the corrosion resistance of the composite coating was enhanced with Ti$_3$SiC$_2$. This could be explained by the following reasons. The FWHM values of the three primary peaks in coating 1 and coating 2 are also listed in Table 4. The FWHM values of the three primary peaks in coating 1 were obviously greater than those of the three primary peaks in coating 2, indicating that the grain size of the $\alpha$-Fe cellular dendrites in coating 1 was smaller than that in coating 2. The larger the grain size, the lower the grain boundaries. Those C atoms adjacent to grain boundaries were prone to interact with Cr to form Cr$_{23}$C$_6$ carbides, thereby decreasing the Cr content at the boundaries and forming Cr-depleted regions at grain boundaries, which resulted in higher activity of grain boundaries during electrochemical measurement. Consequently, grain boundaries and grains tended to form smaller anode or greater cathode microbatteries to accelerate the intergranular corrosion. Thus, it could be concluded that the greater the grain boundaries, the more serious the intergranular corrosion. Furthermore, with the addition of Ti$_3$SiC$_2$, a large number of Ti atoms was released into the molten pool. As the molten pool cooled, TiC and Ti(B,C)$_2$ separated first from the melt pool due to the higher melting point, which reduced the segregation amount of Cr$_{23}$C$_6$ carbides and meant that more free Cr atoms remained in coating 2. Cr was the important element that increased the corrosion resistance of the composite coating. As a result, passivation films rapidly formed on the surface of coating 2, which suppressed the corrosion process. Finally, the corrosion resistance of coating 2 was superior to that of coating 1.

*3.5. Tribological Properties and Mechanisms*

The wear weight loss of the composite coatings after the dry sliding process was measured, as shown in Figure 16. The relationship between the wear weight loss and the testing time showed a positive correlation. The wear weight loss increased with the increase of the testing time. The mean wear mass loss rates of coating 1 and coating 2 were calculated as approximately $1.1 \times 10^{-2}$ mg min$^{-1}$ and $8.5 \times 10^{-3}$ mg min$^{-1}$, respectively, which indicated that the mean wear mass loss rate of coating 1 was nearly twice as high as that of coating 2. Therefore, coating 2 showed higher wear resistance than coating 1 under wear testing conditions.

As shown in Figure 17, many obvious plowing grooves existed on the worn surface of coating 1, coupled with spalling pits. This was because the wear debris fell from the Al$_2$O$_3$ ceramic ball in coating 1, and was transmitted to the secondary abrasive particles under the action of friction force, resulting in plastic deformation and deep plowing grooves. When the plastic deformation reached a certain degree, the nucleation cracks occurred in the subsurface and continued propagating to the surface. Due to the microcutting of the wear debris and crack propagation, the fragments were peeled off from the coating surface. The above facts showed that the predominant wear mechanisms of coating 1 were severe microcutting and fatigue spalling, which were found to be disadvantageous in terms of wear resistance. Compared with coating 1, the worn surface of coating 2 (Figure 17b) was much smoother and some light gray regions also existed, which were probably hard particles. Therefore, the difference of tribological behavior mainly resulted from the existence of reinforcements (TiC, (Cr, Fe)$_{23}$C$_6$, TiC/Al$_2$O$_3$ composite ceramics, Ti(B,C)$_2$, and Fe$_3$C) and self-lubricating Ti$_3$SiC$_2$. Figure 18 shows the SEM images of worn surfaces of the counterparts of Al$_2$O$_3$ balls to elucidate the self-lubricating properties of the composite coatings. The size of the worn surface on the Al$_2$O$_3$ balls in Figure 18a was obviously bigger than that in Figure 18b, suggesting that the negative effect of coating 1 on its counterpart was greater than that of Coating 2. The reason may be that the wear debris that scratched the softer surface of coating 1 may result in the severe three-body abrasive wear and negative

influence on the surface of the counterpart. Additionally, it was found that a dark and continuous transferred film covered the worn surface of the $Al_2O_3$ ball, as shown in Figure 18b, which was helpful in protecting the coating from direct contact and wear from the very hard counterpart $Al_2O_3$ ceramic ball. Thus, it could be concluded that $Ti_3SiC_2$ could be squeezed out from the matrix to form the lubricating film during the dry sliding wear process, which accumulated on worn surfaces of the coating and $Al_2O_3$ ceramic ball, weakening the resistance to shearing. In addition, the microhardness was also closely related to the wear resistance of materials. In general, the higher hardness meant higher wear resistance. As such, mild grooves along the sliding direction were present on the worn surface. Therefore, the wear mechanism of coating 2 was slight microcutting and the formation of the transferred film.

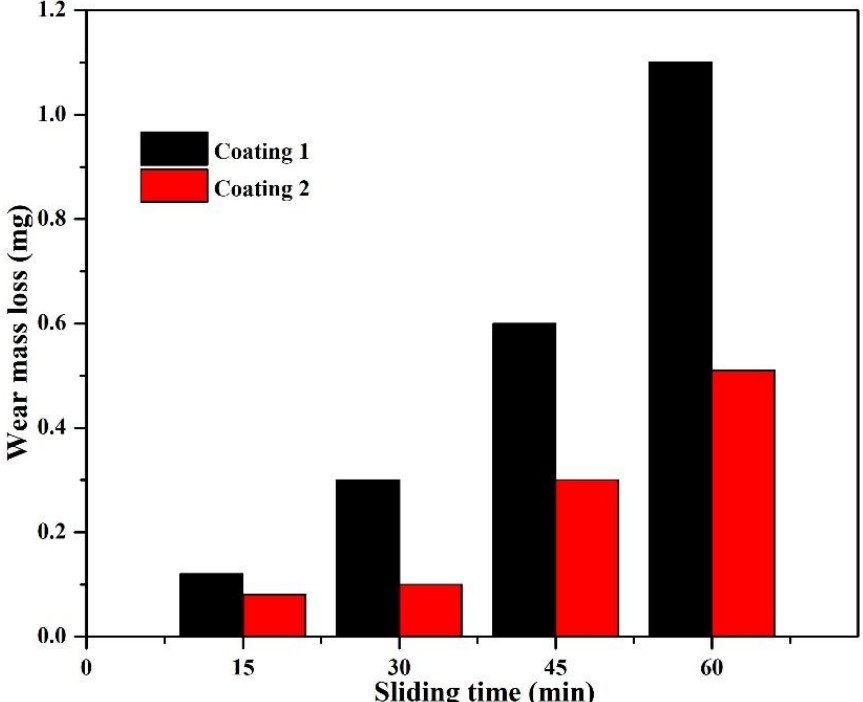

**Figure 16.** Wear mass loss of Coating 1 and Coating 2.

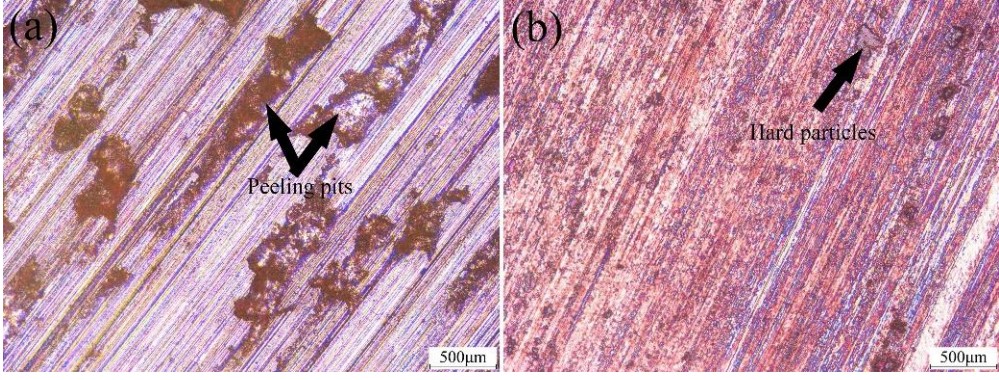

**Figure 17.** Worn surface morphologies of (**a**) coating 1 and (**b**) coating 2.

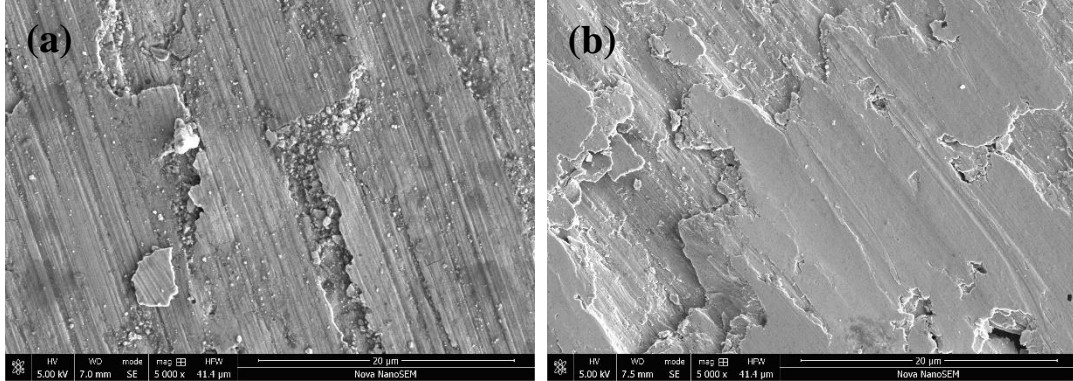

**Figure 18.** SEM morphologies of the worn surfaces of counter-body $Al_2O_3$ balls: (**a**) coating 1; (**b**) coating 2.

## 4. Conclusions

In this article, a Fe-based laser cladding alloy with $Ti_3SiC_2$ was coated on the surface of 16Mn steel. The effects of $Ti_3SiC_2$ on the microstructure and properties of the composite coating were analyzed. The conclusions are as follows:

(1)  With the addition of $Ti_3SiC_2$, many additional reinforcements, such as $(Cr, Fe)_{23}C_6$ carbides, $Ti(B,C)_2$, and $Al_2O_3/TiC$ composite ceramics, were formed in situ in the composite coatings;

(2)  The microhardness, corrosion, and wear resistance of composite coating with 10 wt.% $Ti_3SiC_2$ were about 1.35, 4.3, and 2 times the values of the original coating, respectively, which was mainly attributed to the formation of additional reinforcements;

(3)  Severe fatigue spalling and microcutting were the main wear mechanisms of the original coating, while severe slight microcutting and the formation of the transferred film were the main mechanisms of the composite coating.

**Author Contributions:** Conceptualization, Q.S. and H.Z.; methodology, Q.S. and H.Z.; software, Q.S.; validation, Q.S. and H.Z.; formal analysis, Q.S. and H.Z.; investigation, Q.S. and H.Z.; resources, C.L.; data curation, Q.S., H.Z., and C.L.; writing-review and editing, Q.S., H.Z., and C.L.; supervision, Q.S., H.Z., and C.L.; project administration, Q.S., H.Z., and C.L. All authors have read and agreed to the published version of the manuscript.

**Funding:** This work was financially supported by National Natural Science Foundation of China (51275213, 51302112), the Jiangsu National Nature Science Foundation (BK2011534, BK2011480), and the Scientific and Technological Innovation Plan of Jiangsu Province in China (Grant Nos. CXLX13_645).

**Conflicts of Interest:** The authors declare no conflict of interest.

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
