# Peer review of "The Effects of the Addition of Ti3SiC2 on the Microstructure and Properties of Laser Cladding Composite Coatings"

_coatings, doi:10.3390/coatings10050498_

Round 1
Reviewer 1 Report
The topic of the work and the results carried out are interesting. However, I can consider publishing after the major revision according to the following comments:
- The novelty of the proposed solution in relation to other coatings should be underlined and documented.
- The text of the article is too long, it contains a lot of descriptions of results without specific conclusions and references. Lack of unequivocal explanations of the authors describing the improvement or decrease of coating properties.
- The number of tested coatings seems too small, as a result only two series 0% and 10% were compared, on which basis the concentration of additives was chosen.
- Introduction should be supplemented with literature from recent years (2017-2020).
- Figure captions should be improved with detail, they are incomprehensible: Fig.8; Fig.9; Fig.11;
- Conclusions are too short, do not contain a general summary and specific practical conclusions
Author Response
Responses to Reviewers Dear Reviewers: Thanks very much for taking your time to review our manuscript entitled “Effects of Ti3SiC2 addition on microstructure and properties of laser cladding composite coatings” (Coatings-785139). We really appreciate all your comments and suggestions! Please find my itemized responses in below and my revisions in the resubmitted files. Thanks again! Point 1: The text of the article is too long, it contains a lot of descriptions of results without specific conclusions and references. Lack of unequivocal explanations of the authors describing the improvement or decrease of coating properties. Response 1: We are so grateful for your comments and suggestions. “As presented in Fig.3(a and b), Coating 1 was about 0.9mm thick while Coating 2 had somewhat thinner thick (about 0.6~0.7mm). The thickness of Coating 2 was greatly smaller than that of Coating 1, because of the existence of splashing and decreasing of dilution” was modified into “As presented in Fig.3(a and b), the thickness of Coating 2 (about 0.9mm) was greatly smaller than that of Coating 1 (0.6~0.7mm), because of the existence of splashing and decreasing of dilution”. “Coating 2 compared with Coating 1 also detected some additional diffraction peaks which belonged to TiC, TiB2 and Al2O3” was modified into “Coating 2 compared with Coating 1 also detected some additional diffraction peaks which belonged to TiC, TiB2 and Al2O3. It was because Ti3SiC2 particles dissolved in the molten pool to decompose Ti, Si, C, Al atoms, and Ti, C, Al, O, B in the molten pool combined with each other to generate TiC, TiB2 and Al2O3”. “XPS analysis of the composite coating was also conducted to further characterize the phase constitution, as shown in figure 10. The Ti 2p spectrum contained a wide peak which had a binding energy of about 454 eV, which is characteristic of the Ti-C bonds in TiC. C 1s spectrum also contained a peak at about 284.5 eV, which may be associated with TiC. The O 2p spectrum contained a wide peak which had a binding energy of about 531 eV, which is characteristic of the Al-O bonds in Al2O3. Al 2p spectrum also contained a peak at about 74.2 eV, which may be associated with Al2O3. Thus, Al2O3 and TiC were detected in XPS. The Gibbs free energy (ΔG0) of Al2O3 was far lower than that of TiC by comparing the two curves in Fig.5b, so Al2O3 would be more likely to precipitate earlier than TiC from the liquid. As soon as Al2O3 particles formed, Ti and C atoms would diffuse towards the early precipitated Al2O3 particles, which had a high surface energy to reduce the nucleation work to form TiC. Fig.11 shows schematic diagram of nucleation and growth behaviors of in-situ synthesized Al2O3/TiC composite ceramics. Al2O3/TiC composite ceramics accumulated at grain boundaries, which could hinder the movement of grain boundaries in order to increase the wear resistance of the composite coatings.” Was added to the manuscript. Thank you once again. Point 2: The number of tested coatings seems too small, as a result only two series 0% and 10% were compared, on which basis the concentration of additives was chosen. Response 2: We are so grateful for your valuable comments and suggestions. Fe60 alloy powders with 10 wt.% can be successfully deposited on the 16Mn steel substrate. However, according to previous experiments, with the content of Ti3SiC2 increasing to 15wt.%, coating powders occurred severe splashing during the laser cladding process, causing the thickness of composite coating extremely thin, as shown in the following figure. Thus, with 15wt.% Ti3SiC2 addition, the prepared coating had little effect on the protection of the substrate. Thus, the content of Ti3SiC2 in the composite coating reached the maximum value 10wt.%. Thank you once again. Figure the morphology of Fe-based alloy/15wt.% Ti3SiC2 coating. Point 3: Introduction should be supplemented with literature from recent years (2017-2020). Response 3: We are so grateful for your valuable comments and suggestions. “5. S. Yang, N. Chen, W.J. Liu, M.L. Zhong, Z.J. Wang, Hiroyuki Kakawa. Surf. Coat. Technol. 2004, 183, 254–262. 6. J.J. Candela, J.A. Jimenez, P. Franconetti, V. Amigó. J. Mater. Process. Technol. 2014, 214, 2325–2332. 7. Wang Y, Zhao S, Gao W. J. Mater. Process. Technol. 2014, 214(4), 899–905” was modified into “5. J. Zeisig, N. Schädlich, L. Giebeler. Wear 2017, 382–383:107-112. 6. Kaiming Wang, Dong Du, Guan Liu, Baohua Chang. Science and Technology of Welding and Joining, 2019, 24: 517-524. 7. X.Li, C.H. Zhang, S.Zhang. Optics and Laser Technology 2019, 114:209–215. 8. Y.Y. Sui, F. Yang, G.L. Qin, Z.Y. Ao, Y. Liu, Y.B. Wang. J. Mater. Process. Technol. 2018, 252:217–224. 9. L.L. Zhai, C.Y. Ban, J.W. Zhang. Surface Coat. Technol. 2019, 358 (25):531–538. 10. L.L. Zhai, C.Y. Ban, J.W. Zhang. Optics and Laser Technology 2019,114: 81–88.”. Thank you once again. Point 4: Figure captions should be improved with detail, they are incomprehensible: Fig.8; Fig.9; Fig.11; Response 4: We are so grateful for your valuable comments. “Figure 8. The magnified morphologies of the red rectangles B (a), C (b) and D (d). Figure 9. Linear elemental distributions of “eyeball” structure. Figure 11. The elemental distributions of irregular cubic blocks” was modified into “Figure 8. Enlarged views of red rectangles: (a) area B in figure 7a; (b) area C in figure 7a; area D in figure 7b. Figure 9. Linear elemental distributions of eyeball-like phases in figure 8a. Figure 11. Elemental distributions of irregular cubic blocks in figure 8b.”. Thank you once again. Point 5: Conclusions are too short, do not contain a general summary and specific practical conclusions. Response 5: We are so grateful for your valuable comments and suggestions. Conclusions have been modified carefully. The new conclusions as follows: “In this article, Fe-based alloy coatings with Ti3SiC2 were successfully fabricated on the surface of 16Mn steel by laser cladding. And effects of Ti3SiC2 on microstructure and properties of the composite coating were conducted. The following conclusions could be drawn: (1) With Ti3SiC2 addition, many additional reinforcements were in-situ formed in the composite coatings, such as TiC, eyeball-like Al2O3/TiC composite ceramics, Ti(B,C)2 and (Cr, Fe)23C6 carbides. (2) The microhardness, corrosion and wear resistance of composite coating with 10 wt.%Ti3SiC2 were about 1.35, 4.3 and 2 times that of original coating, respectively, which was mainly attributed to the formation of additional reinforcements. (3) The main wear mechanism of original coating were severe micro-cutting and fatigue spalling, and the main mechanism of composite coating with Ti3SiC2 were severe slight micro-cutting and the formation of the transferred film” was added to the manuscript. Thank you once again. Finally, we appreciate for your warm work earnestly, and hope that the correction above will meet with approval. Once again, thank you very much for your comments and suggestions. Yours sincerely, Qin Shi

Reviewer 2 Report
The authors have shown positive effects of Ti3SiC2 addition in a composite coating on steel substrate. The content is good, however following comments (see the attached pdf with comments and texts highlighted in yellow) need to be addressed before considering for publication.

Author Response
Dear Reviewers:
Thanks very much for taking your time to review our manuscript entitled “Effects of Ti3SiC2 addition on microstructure and properties of laser cladding composite coatings” (Coatings-785139). We really appreciate all your comments and suggestions! Please find my itemized responses in below and my revisions in the resubmitted files.
Thanks again!
Point 1: Are both coatings deposited with same laser parameters? Are the parameters optimized for these two coatings?
Response 1: We are so grateful for your valuable comments and suggestions. Yes, these experimental parameters are verified by a large number of experiments, and finally found that the coatings prepared by this parameter have the highest quality. Thank you once again.
Point 2: Are both coatings deposited with same laser parameters? Are the parameters optimized for these two coatings?
Response 2: We are so grateful for your valuable comments and suggestions. The scales have been modified carefully. Thank you once again.
Figure 3. Macrograph of transverse cross-section of Coating 1 and Coating 2: (a) overview of Coating 1, (b) overview of Coating 2, (c) bonding area of Coating 1 and (d) bonding area of Coating 2.
Point 3: if this these are optical microprograms, then why SEM, Zeiss is mentioned?
Response 3: We are so grateful for your valuable comments and suggestions. The scales have been modified carefully. This is really my mistakes. The phase constituent and microstructure were identified with the assistance of X-ray diffractometer (XRD-6000), scanning electron microscope (ZEISS Merlin Compact) and energy dispersive spectrometer (EDS). However, the worn surfaces of composite coatings were observed by optical electron microscopy, which was used to investigate their wear mechanism. Thus, “optical electron microscopy (SEM, ZEISS)” in the manuscript was modified into “optical electron microscopy”. Thank you once again.
Point 4: please identify coating and substate in the image for convenience of the reader.
It is difficult to understand the presence of Ti3SiC2 in the coating from the EDS spectra. Instead of putting all the elements, only Ti, Si, C, and Fe would be.
Response 4: We are so grateful for your valuable comments and suggestions. Coating and substate in the image have been identified. EDS spectra can not prove the existence of Ti3SiC2 in the coating. It is observed from EDS spectra that the melted alloy atoms at the interface between the clad coatings and substrate appeared mutual diffusion, suggesting that high quality clad coatings were well adherent to the substrate. The presence of Ti3SiC2 in the coating can be detected from XRD. Thank you once again.
Point 5: can you please mention what are 'some' chemical reactions that affects the coating colour?
Response 5: We are so grateful for your valuable comments and suggestions. Chemical reactions probably occur as follows:
Point 6: do you mean you to say that coating 2 is less thick because of Ti3SiC2 addition? why? please rearrange the sentence.
Response 6: We are so grateful for your valuable comments and suggestions. “As presented in Fig.3a and b, Coating 1 was about 0.9mm thick while Coating 2 had somewhat thinner thick (about 0.6~0.7mm). The thickness of Coating 2 was greatly smaller than that of Coating 1, because of the existence of splashing and decreasing of dilution. Consequently, the addition of Ti3SiC2 during laser cladding resulted in the thickness of coating.”was modified into “As presented in Fig.3(a and b), the thickness of Coating 2 (0.6~0.7mm) was greatly smaller than that of Coating 1(about 0.9mm). It was because that the addition of Ti3SiC2 led to powders’ splashing laser cladding, finally, decreasing dilution. ” . Thank you once again.
Point 7: please identify what is meant by intermediate region in respect to the coating thickness. why no images from bottom and upper regions are shown and discussed for coating 1 as it has been done for coating 2?
Response 7: We are so grateful for your valuable comments and suggestions. it is really my mistakes. The image shows the upper region of Coating 1. We have modified it carefully. Thank you once again.
Point 8: The Ti3SiC2 peaks are observed along with many other peaks, such as TiC, Al2O3 and TiB2. Moreover, the cumulative peak intensity is very low. Thus, it is difficult to conclude that if those little peaks are due to Ti3SiC2 un-melted particles.
Response 8: We are so grateful for your valuable comments and suggestions. However, in XRD, if a substance can detect its three strong peaks, we can identify its existence. Thus, in this XRD pattern, three strong peaks of Ti3SiC2 can be detected, thus, it can be concluded that a small amount of Ti3SiC2 reserved in the composite coatings after laser cladding. Thank you once again.
Point 9: please identify what is meant by bottom, intermediate and upper regions in respect to the coating thickness.
please identify B, C, D as shown within the images or somewhere mention that these are shown later in image 8.
Response 9: We are so grateful for your valuable comments and suggestions. One-third of the thickness of the coating from the substrate is the bottom, and One-third of the thickness of the coating from the surface of coating. Others is the intermediate region of the coating. Figure 7 shows B, C, D in the image. Thank you once again.
Point 10: From fig. 9, only the changes from Ti, Fe and Cr spectra can be detected, not the others. All other elements show almost no visible changes from matrix to the eye-ball like phases. Thus clear evidence is needed to say this.
Same comment as before: EDS spectra from Fig. 9 does not show visible changes in Al, O, C contents from the matrix to the eyeball. Thus additional evidences are needed to claim if the core is rich with Al and O, while outside is rich with C. Also, EDS is not efficient to detect light materials such as O and C, particularly when C is below 1 wt%.
Response 10: We are so grateful for your valuable comments and suggestions. XPS analysis of the composite coating was also conducted to further characterize the phase constitution, as shown in figure 10. The Ti 2p spectrum contained a wide peak which had a binding energy of about 454 eV, which is characteristic of the Ti-C bonds in TiC. C 1s spectrum also contained a peak at about 284.5 eV, which may be associated with TiC. The O 2p spectrum contained a wide peak which had a binding energy of about 531 eV, which is characteristic of the Al-O bonds in Al2O3. Al 2p spectrum also contained a peak at about 74.2 eV, which may be associated with Al2O3. Thus, Al2O3 and TiC were detected in XPS. Thus, clear evidence has been provided. Thank you once again.
Figure 10. XPS element valence diagram of the composite coating with Ti3SiC2: (a) Ti 2p; (b) C 1s; (c) O 1s; (d) Al 2p.
Point 11: Recheck this sentence, as the description does not match with the elemental analysis shown in Fig. 12. By the way, Fig 12 does not show any presence of B, although it has been written that the hexagonal phases are consisted of B.
Response 11: We are so grateful for your valuable comments and suggestions. Fig.12 has been replaced by new images in the manuscript. As seen, the hexagonal phases contained B, C, Ti elements. In addition, the content of B is higher than that of C.
Figure 12. The elemental distributions of area H in Fig.10c.
Point 12: why coating 2 shows a drastic increase in the wear rate after 30 min of sliding, whereas coating 1 shows linearly increasing wear rate since the beginning?
Response 12: We are so grateful for your valuable comments and suggestions. Coating 2 showed a drastic increase in the wear rate after 30 min of sliding, whereas coating 1 showed linearly increasing wear rate since the beginning. It was because that the wear resistance of composite coatings was related to the microstructure of composite coatings. Many hard phases in Coating 2 could be peeled out from the matrix when Coating 2 subjected to longer period of wear, which led to a drastic increase in the wear rate. The microstructure of Coating 1 was uniform, thus, coating 1 showed linearly increasing wear rate since the beginning.
Point 13: These are optical images, not SEM images as authors have already said in section 2.2, line 115.
Response 13: We are so grateful for your valuable comments and suggestions. The worn surfaces of composite coatings were indeed observed by optical electron microscopy, which was used to investigate their wear mechanism.
Point 14: Optical images, not SEM images, please correct.
|
|
Response 14: We are so grateful for your valuable comments and suggestions. Optical images have been corrected carefully. Thank you once again.
Figure 18. SEM morphologies of the worn surfaces of counter-body Al2O3 balls: (a, b) Coating 1 and (c, d) Coating 2(c, d).
Finally, we appreciate for your warm work earnestly, and hope that the correction above will meet with approval.
Once again, thank you very much for your comments and suggestions.
Yours sincerely,
Qin Shi

Reviewer 3 Report
The development of titanium-based coatings is of great interest for metallic compounds. Due to its exceptional properties, enhancements can be prominent, for example regarding wear resistance. However its deposition, as many other types of coatings, may be difficult leading to detachment between layers.
This manuscript focuses on the ‘Effects of Ti3SiC2 addition on microstructure and properties of laser cladding composite coatings’. It explores the effect of the addition on the microstructure and properties of the Fe60 coating by using X-ray diffractometer (XRD), scanning electron microcopy (SEM) coupled with energy dispersion spectroscopy (EDS), Vicker’s indentation, electrochemical and wear tests. The manuscript is globally well structured, easy to read and may be of interest to the readers of the journal. The first part ‘Introduction’ gives a concise background about laser cladding and Ti3SiC2 based coatings before stating the goal of the manuscript. The second part presents the experimental procedures. Some minor additions should be made in this part in my opinion. The third part clear exposes the various results. Finally the conclusion part summarizes the main results but lacks some perspectives.
The minor remarks can be summarized as follows:
- please place the figures and tables after their first evocation in the text (fig 1, 2, …)
- Table 1: indicate the measurement methodology
- Fig 1: precise the diameter estimation (direct usage of SEM?)
- l 69: please replace ‘stand paper’ by sandpaper
- fig 5: please increase the size of the graphs as legends can be hardly visible
Author Response
Dear Reviewers:
Thanks very much for taking your time to review our manuscript entitled “Effects of Ti3SiC2 addition on microstructure and properties of laser cladding composite coatings” (Coatings-785139). We really appreciate all your comments and suggestions! Please find my itemized responses in below and my revisions in the resubmitted files.
Thanks again!
Point 1: please place the figures and tables after their first evocation in the text (fig 1, 2, …)
Response 1: We are so grateful for your valuable comments and suggestions. The figures and tables after their first evocation in the text have been placed carefully, which was shown in the manuscript. Thank you once again.
Point 2: Table 1: indicate the measurement methodology
Response 2: We are so grateful for your valuable comments and suggestions. The chemical composition of 16Mn substrate and Fe60 alloy powder (in wt.%) were measured by EDS. Thus, “Table 1. The chemical composition of 16Mn substrate and Fe60 alloy powder (in wt.%)” was modified into “Table 1. The chemical composition of 16Mn substrate and Fe60 alloy powder (in wt.%) measured by EDS”. Thank you once again.
Point 3: Fig 1: precise the diameter estimation (direct usage of SEM?).
Response 3: We are so grateful for your valuable comments and suggestions. The diameter was based on the diameter given by the company of origin, and it was directly measured by SEM. Thank you once again.
Point 4: please replace ‘stand paper’ by sandpaper.
Response 4: We are so grateful for your valuable comments and suggestions. “stand paper” has been modified into “sandpaper”. Thank you once again.
Point 5: fig 5: please increase the size of the graphs as legends can be hardly visible.
Response 5: We are so grateful for your valuable comments and suggestions. The size of the graphs has been increased. Thank you once again.
Finally, we appreciate for your warm work earnestly, and hope that the correction above will meet with approval.
Once again, thank you very much for your comments and suggestions.
Yours sincerely,
Qin Shi
Round 2
Reviewer 1 Report
The authors have made the required corrections.
Now the article is suitable for publication.
Reviewer 2 Report
The comments are properly addressed.